# Adsorption of Metformin on Activated Carbon Produced from the Water Hyacinth Biowaste Using $H_3PO_4$ as a Chemical Activator

**Ahmad Hakky Mohammad [1], Ivona Radovic [1], Marija Ivanović [2] and Mirjana Kijevčanin [1,\*]**

[1]  Faculty of Technology and Metallurgy, University of Belgrade, Karnegijeva 4, 11120 Belgrade, Serbia
[2]  Vinča Institute of Nuclear Sciences, National Institute of the Republic of Serbia, Department of Materials, University of Belgrade, Mike Petrovića Alasa 12-14, Vinča, 11000 Belgrade, Serbia
\*  Correspondence: mirjana@tmf.bg.ac.rs; Tel.: +381-11-3370-523

**Abstract:** Water hyacinth biomass was used for the synthesis of activated carbons in the process of chemical activation with $H_3PO_4$, followed by controlled carbonization. The study investigates the effect of various impregnation weight ratios of $H_3PO_4$ and dry hyacinth (0.5–3.0), as well as different carbonization temperatures (T = 400–800 °C), on the surface characteristics of the produced activated carbons (AC). The activated carbon obtained with an impregnation ratio of 1.5 and a carbonization temperature of 600 °C (1.5 AC/600) showed the highest values of specific surface area of 1421 $m^2$ $g^{-1}$, representing a selected adsorbent for metformin removal. The chosen sample was characterized by elemental analysis, adsorption–desorption isotherms of nitrogen at −196 °C, via FTIR spectroscopy and the SEM method. The modeling of the experimental adsorption data showed that metformin adsorption: (i) can be best described by the Langmuir isotherm model, with the value of $q_{max}$ = 122.47 mg $g^{-1}$; (ii) led the pseudo-second order kinetic model; and (iii) is a spontaneous ($\Delta G°$ = −3.44 kJ $mol^{-1}$) and endothermic ($\Delta H°$ = 8.77 kJ $mol^{-1}$) process. A desorption study has shown that 92% of metformin was successfully desorbed in the presence of a 0.1 MHCl/ethanol mixture (volume ratio 2:1). The recovery of the adsorbent of 84%, after five successive cycles, indicated that the 1.5 AC/600 has potential to be applied in the real systems for water treatment.

**Keywords:** water hyacinth biowaste; activated carbon; metformin hydrochloride adsorption; isotherm; kinetics

## 1. Introduction

In recent years, increasing consumption of pharmaceutics and drugs has influenced harmful effects on the environment and living world. Metformin has become one of the most consummated anti-diabetic drugs for diabetes mellitus (type 2) and represents one of the most abundant pharmaceuticals being introduced into the environment [1]. A complete and detailed study related to the impact of this drug on the aquatic and soil eco-systems is still missing. However, several hazardous effects caused by the presence of metformin in surface waters have been reported [1]. Despite the treatment of communal water before discharge, the presence of pharmaceutics in surface waters constantly grows, since conventional wastewater treatment is not always designed to eliminate a wide range of different organic substances along with other inorganic and biological contaminants [2]. Therefore, it seems that finding an appropriate method for the removal of all metformin and pharmaceutics should be an urgent task for scientists all over the world.

Several physical and biochemical methods have been employed in order to remove pharmaceuticals from wastewater, such as oxidative catalytic reactions [3], adsorption [2], treatment by membrane bioreactors [4], recovery by bio-systems [5], etc. Among the applied methods, adsorption has been proven as a high-quality technique for the removal of dissolved organic pollutants from industrial and municipal wastewater. The adsorption

method has been widely applied for this purpose, since it is easy to handle and is also a highly efficient and low-cost method. However, the great challenge presented is finding good and suitable adsorbents, capable of adsorbing a wide range of pollutants.

Activated carbons as adsorbents that have received a lot of attention over the decades, due to their unique surface characteristics, i.e., the presence of different functional groups responsible for various adsorption mechanisms as well as highly developed surface micro- and meso-porosity [6,7]. On the other hand, the cost of a commercial activated carbon is high, if its production is based on non-renewable and expensive fossil materials such as coal [8]. Low-cost activated carbons can be obtained from different lignocellulose materials, as well as other industrial and agricultural by-products [6–8].

Water hyacinth (*Eichhornia crassipes*) is a floating water plant found in rivers and lakes. Although, originally, it has been used as an ornamental plant, nowadays water hyacinth represents a problematic weed due to its rapid spread. The floating mats of water hyacinth cause significant economic and environmental problems [9,10]. However, due to its favorable chemical composition, water hyacinth could be used as a precursor for activated carbons production. The contents of the lignocelluloses (60% cellulose, 8% hemicelluloses, and 17% lignin) and a high carbon–nitrogen ratio make water hyacinth a suitable raw material for the carbonization process and a source of alternative renewable energy [11,12].

The chemical activation of raw lignocellulose material represents a necessary step in the process of activated carbons production. Many chemicals can be used as active agents in the chemical activation of the lignocelluloses' precursors: potassium hydroxide, sodium hydroxide, sodium carbonate, zinc-chloride, phosphoric acid, etc. [13,14]. Among the various activating agents used for activated carbons production, it seems that $ZnCl_2$, KOH, and $H_3PO_4$ have been most frequently applied due to their efficiency in the process of porous structure development. Besides a strong contribution to the pore development by localized decomposition of organic matters, $ZnCl_2$ also shows a high activating capability. However, $ZnCl_2$ is relatively costly and may cause corrosion, plus the activated carbons obtained by using $ZnCl_2$ cannot be used in pharmaceutical and food industries since they can contaminate the product [14,15]. Potassium hydroxide has been widely used in the production of activated carbons with highly developed microporosity. The most prominent advantages of KOH usage in the manufacturing of activated carbons are low cost, a high surface area, formation of the hydroxyl functional groups on the carbon surface, and carbonization temperatures up to 700 °C. However, the application of KOH causes considerable concern due to its toxicity and detrimental, toxic impacts on humans and the environment [16]. Therefore, phosphoric acid seems to show some advantages related to the environmental impact [14,17] and the possibility of the application of activated carbons in the food and pharmaceutical industry [15]. Further, phosphoric acid led to the production of activated carbons that developed both microporous and mesoporous surfaces [17].

The main objective of the present work was to obtain activated carbon with a high surface area from water hyacinth biomass, using various impregnation ratios of phosphoric acid as a chemical activator and applying different carbonization temperatures. According to the authors' best knowledge, there is no literature study where the effects of both synthesis parameters on the surface properties of activated carbons were investigated. The activated carbon with the highest value of the specific surface area was selected and tested as an adsorbent of a pharmaceutic—metformin. An adsorption study revealed the effects of the initial metformin concentration, the adsorbent concentration, and the pH and temperature of the adsorbate solution on the amount and rate of the adsorbed metformin. Different adsorption isotherm models including Langmuir, Freundlich, and Redlich–Paterson were employed to describe the adsorption data. The appropriate kinetics models of the pseudo-first order, pseudo-second order, and intra-particle diffusion were used to study the kinetics of the adsorption. A thermodynamic study was performed in order to determinate the values of the $\Delta G°$, $\Delta H°$ and $\Delta S°$ of the investigated process. In

this study, the activated carbon produced from water hyacinth has been evaluated, for the first time, as an adsorbent of metformin.

## 2. Materials and Methods

### 2.1. Materials

The water hyacinth (WH) plant (Karbala, Iraq) was used as a raw material for activated carbons synthesis. The raw WH was washed with distilled water in order to remove mud and dust. The leaves of the plant were separated, and roots and stalks were chopped and dried in an oven for 24 h. Further, the raw material was boiled in 0.25 M hydrochloric acid, rinsed well with distilled water until negative reaction for $Cl^-$ ions (test with 0.1 M $AgNO_3$), and dried in a vacuum freeze dryer for 24 h. Finally, dry WH has been ground and sieved in order to obtain particles less than 2.0 mm [18].

The phosphoric acid used for the impregnation of dry WH material was supplied from Sigma Aldrich, with purity of 85% (*w/v*).

The anti-hyperglycemic drug metformin hydrochloride (MTF), supplied by Sigma Aldrich, CAS-No. 1115-70-4, with purity $\leq$ 100%, was used as a model of pharmaceutic pollutant.

### 2.2. Preparation of the Activated Carbons

Preparation of the activated carbon from the dry WH was carried out by a chemical activation process with the $H_3PO_4$ activating agent. The impregnation ratio was determined as the ratio of the weight of $H_3PO_4$ to the weight of the dried WH. The dry and ground WH was impregnated with an $H_3PO_4$ solution in 0.5 to 3.0 ratios, with an increment of 0.5. The 20 g of dry WH sample was added to 80 mL of solution, with an appropriate mass of $H_3PO_4$, and stirred at 60 °C for 4 h. The solid and liquid phases were separated by filtration through a Buchner funnel and dried at 105 °C for 24 h. The carbonization of activated WH was carried out in an electrical furnace with nitrogen flowing (150 $cm^3$ $min^{-1}$) and a heating rate of 15 °C $min^{-1}$. Carbonization for 80 min was conducted at the following temperatures: 400 °C, 500 °C, 600 °C, 700 °C, and 800 °C. The obtained activated carbons were rinsed with hot distilled water until at neutral pH and finally dried at 110 °C for 12 h. The synthesized activated carbons were denoted according to the impregnation ratio and carbonization temperature, e.g., 1.0/600C means that impregnation ratio and carbonization temperature were 1.0 and 600 °C, respectively. A schematic presentation of the raw water hyacinth preparation and activated carbons synthesis is given in Figure 1.

The yield of activated carbons was calculated from the mass ratio between activated carbon and starting WH after drying process, as follows:

$$Y = (m_{AC}/m_{dryWH}) \times 100\% \tag{1}$$

where $Y$ (%) is the yield of the synthesis, $m_{AC}$ (g) is the mass of activated carbon, and $m_{dryWH}$ (g) is the mass of dry WH.

### 2.3. Methods of Characterization

The synthetized ACs were characterized using elemental analysis, nitrogen adsorption–desorption isotherms at −196 °C, FTIR spectroscopy, and scanning electron microscopy (SEM).

The elemental analysis was used in order to determine the content of carbon, hydrogen, and nitrogen in raw material and activated carbons. The elemental analysis was performed using elemental analyzer instrument (Thermo Scientific -FlashEA1112 Automatic Elemental Analyzers, Waltham, MA, USA).

The textural properties of ACs were obtained from the adsorption–desorption isotherms of nitrogen at −196 °C (Micromeritics' ASAP 2020). Prior to the analysis, the samples were outgassed at 110 °C for 10 h. The specific surface area ($S_{BET}$) was calculated according to the Brunner–Emmett–Teller method [19], and the total pore volume ($V_T$) was estimated from the $N_2$ adsorption isotherm according to Gurvich rule, which represents the liquid

molar volume adsorbed at pressure $p/p_0$ of 0.999 [19,20]. The volume of micropores was calculated using the Dubinin–Radushkevich method [20], while values of mesopore volumes were obtained according to the Barrett, Joyner, Halenda (BJH) method [21]. The pore size distribution of the prepared activated carbon was determined by the BJH model, and the mean pore diameter ($D_P$) was calculated from $D_P = 4V_T/S_{BET}$ [13].

**Figure 1.** Schematic presentations of the raw WH preparation and activated carbons synthesis.

FTIR spectroscopy was used in order to analyze the presence of the surface-active functional groups (Thermo Nicolet iS 5 FTIR). The KBr pastille was prepared by mixing 2 mg of activated carbon sample with 200 mg of KBr. The spectrum was recorded in wave-numbers range from 400 and 4000 cm$^{-1}$.

The morphology of the selected activated carbon was characterized by scanning electron microscopy (SEM-JEOL, JSM 6360 LV, JEOL, Tokyo, Japan).

Determination of zero-point charge (PZC) was performed in order to estimate the effect of the pH of adsorbate solution on the surface charge of selected activated carbon. The measure of the $pH_{PZC}$ was conducted according to the well-known procedure, as previously described in the literature [22]. The 10 mM solution of NaCl was adjusted between 2 and 12 using HCl or NaOH solution (0.1 M). The 100 mg of activated carbon was added to the 50 cm$^3$ of NaCl solution, and the suspension was mixed at room temperature for 48 h. After the separation of the solid phase by centrifugation, the final supernatant pH value was measured. The intersection point for curve $pH_{final}$ vs. $pH_{initial}$ and the line that represents the bisector have been taken as $pH_{PZC}$.

*2.4. Adsorption—Desorption Experiments*

Batch adsorption studies were carried out in a thermostatic shaker (KS 4000I Control Shaker 115V IKA 3510001, IKA®-Werke GmbH & Co., KG, Staufen, Germany). After the adsorption process, the MTF concentration in the solution was analyzed by spectrophotometer (UV-VIS 1800, Shimadzu, Kyoto, Japan) at $\lambda_{max}$ = 234 nm, with the previous separation of the solid phase by centrifugation (Hettich Universal 320, Hettich GmbH & Co., KG, Tuttlingen, Germany) at 17,000 rpm for 15 min. The percentage of the removal ($R$) of MTF from aqueous solutions was determined using the following equation:

$$R = ((C_0 - C_e)/C_0) \times 100\% \tag{2}$$

The amount of the adsorbed MTF ($q_t$, mg g$^{-1}$) in time $t$ per mass of adsorbent was calculated according to the following equation:

$$q_t = (C_0 - C_e) \times V/m_{ads} \tag{3}$$

where $C_0$ and $C_e$ are initial MTF concentrations and MTF concentration after adsorption time $t$ (mg L$^{-1}$), respectively, $V$ is the volume (mL) of the MTF solution, and $m_{ads}$ is the adsorbent mass (mg).

In preliminary experiments, linearity between the absorbance of MTF standard solutions (concentration range 1–20 mg L$^{-1}$) and absorbance was determined according to the Beer–Lambert law. MTF solutions with a concentration above 20 mg L$^{-1}$ were diluted prior to measurement.

The effect of the adsorbent mass, adsorbate initial concentration, pH, and temperature on the adsorption of MTF were investigated. The impact of the adsorbent mass on the amount of removed MTF was tested for 120 min in the range of 100 mg L$^{-1}$ to 500 mg L$^{-1}$, using MTF solution with initial concentration of 20 mg mL$^{-1}$. To study the effect of the initial MTF concentration on the adsorption, experiments were carried out at the temperature of 25 °C, using a mass concentration of the adsorbent of 250 mg L$^{-1}$ and an initial MTF concentration of 10, 20, 40, 60, 80, and 100 mg L$^{-1}$. The effect of pH was tested in the pH range from 2.0 to 12.0, while all the other parameters were kept constant (initial MTF concentration of 20 mg L$^{-1}$ and adsorbent concentration of 250 mg L$^{-1}$). The pH was adjusted using 0.1 mol L$^{-1}$ NaOH and 0.1 mol L$^{-1}$ HCl solutions. Finally, the dependence of the amount of adsorbed MTF on temperature was investigated in temperature range from 25 to 50 °C, keeping other adsorption parameters constant, as previously described.

Desorption studies were conducted to select the optimum desorbing solution to be employed for adsorbent regeneration. Prior to desorption, the saturation of adsorbent was performed from MTF solution, according to the following procedure. The 100 mg of adsorbent was saturated using 400.00 mL of 100 mg L$^{-1}$ MTF solution in an equilibrium time of 120 min at 25 °C. After saturation, the MTF-saturated adsorbent was separated from solution by centrifugation, rinsed with distilled water, and dried under the vacuum at room temperature. The desorption was primarily performed from 0.1 M NaOH, 0.1 M HCl, and 0.1 M NaCl, and then from mixtures of 0.1 M HCl and 96% ethanol in defined volume ratios. During the desorption experiment, the MTF-saturated adsorbent was constantly shaken for 2 h at room temperature. The concentration of desorbed MTF in the supernatant was measured, while the percentage of desorption was calculated using Equation (4):

$$\text{Percentage of desorption (\%)} = (C_{des}/C_{ads}) \times 100\% \tag{4}$$

where $C_{ads}$ and $C_{des}$ denote the amounts of adsorbed and desorbed MTF, respectively.

After MTF desorption and separation, the recovered adsorbent was washed with distilled water, dried, and reused in the next adsorption cycle.

## 3. Results and Discussions

### 3.1. Results of the Characterization

3.1.1. Yield of the Activated Carbons

The yields of activated carbons obtained after chemical activation with $H_3PO_4$ and the carbonization process in the temperature range from 400–800 °C are given in Figure 2.

Figure 2 shows that yields for all samples of activated carbon were in the range of 17.8% to 39.8%. These values are comparable to those obtained in other studies [17,23]. The activated carbons obtained without activation with $H_3PO_4$ had relatively low yields, in the range of 17.8% to 24.9%. This fact has been discussed in our previous publication [18] as well as in the literature [23]. The low yields of activated carbons, with impregnation ratio 0, are related to significant weight loss attributed to the extraction of different gas compounds: CO, $CO_2$, and $CH_4$ [23]. Generally, the best yields were obtained for a carbonization temperature of 400 °C. With temperature increase, the yield of activated carbon decreased, regardless the impregnation ratio. This fact has been already explained by the promotion of tar volatilization at higher temperatures [17,24].

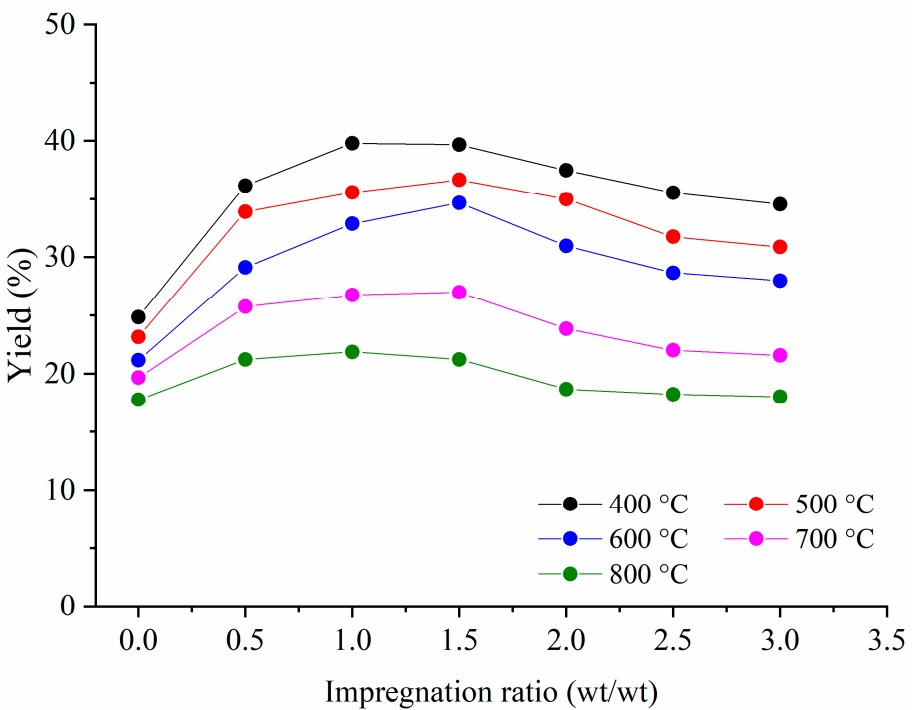

**Figure 2.** The influence of impregnation ratio with $H_3PO_4$ and carbonization temperature on the yield of the produced activated carbons.

3.1.2. Effect of the Activation Agent and Carbonization Temperature on Surface Characteristics

The amount of the activation agent has a similar impact on yields for each carbonization temperature, i.e., activated carbon yields increase as the impregnation ratio rises up to 1.0 or 1.5.

For carbonization temperatures of 500 °C and 600 °C, the highest yields were obtained for an impregnation ratio of 1.5, while for other carbonization temperatures, the highest yield was achieved with an impregnation ratio of 1.0. Generally, there is no significant difference in yields between these two values of impregnation ratios. The higher amount of phosphoric acid (above 1.5) leads to a continuous yields decrease. This trend can be attributed to the specific role of phosphoric acid in the activation process. First, the yields of activated carbons increase, since the phosphoric acid promotes degradation processes such as depolymerization, dehydration, and redistribution of lignocellulose materials as well as conversion of aliphatic compounds to aromatic compounds [14]. Phosphoric acid also reacts with char, which leads to the formation of volatile compounds. With higher amounts of phosphoric acid, this process takes a significant role, leading to an increase in weight loss and, consequently, to a low yield of activated carbons [14].

The effects of the impregnation ratio and applied carbonization temperature on the specific surface area ($S_{BET}$) of activated carbons are shown in Figure 3.

Figure 3a shows that the activated carbon obtained without impregnation exhibited the lowest value of $S_{BET}$, which was expected, since the activation agent had a significant role in the surface development during the carbonization process [14]. The introduction of phosphoric acid led to an increase in the specific surface area, upon increasing the impregnation ratio from 0 to 1.5. After a further impregnation ratio increase from 1.5 to 3, the surface area changes can be considered approximately constant. Based on this result, the samples with an impregnation ratio of 1.5 were selected for further analysis.

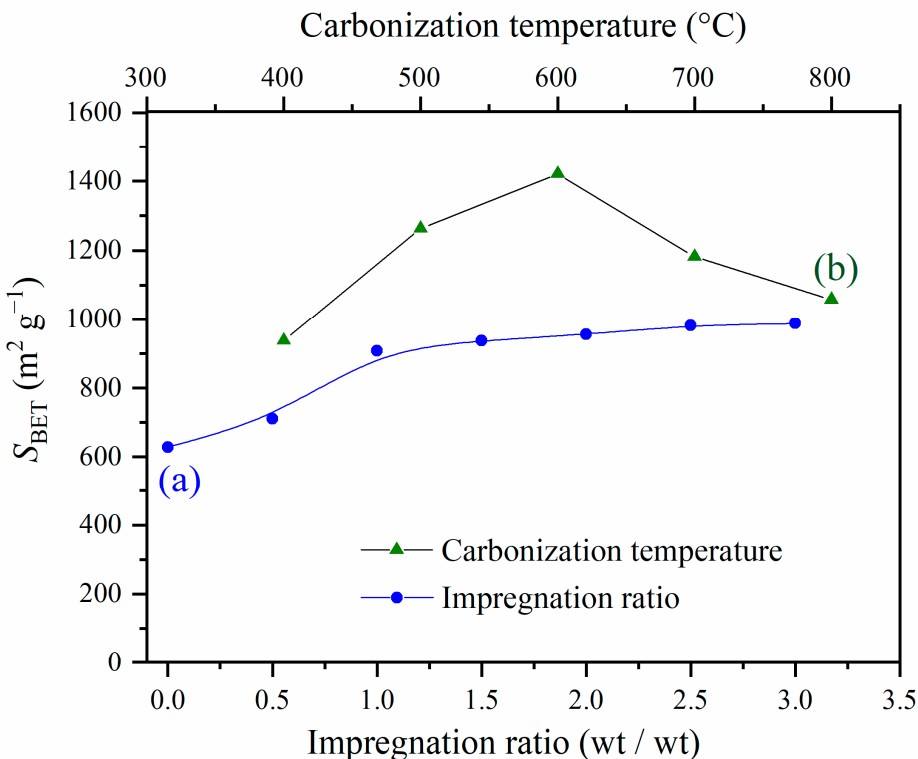

**Figure 3.** (**a**) The effect of the impregnation ratio on the specific surface area of activated carbons obtained by carbonization at 400 °C; (**b**) the effect of the carbonization temperature on the specific surface area of activated carbons with the impregnation ratio 1.5.

It is well-known that carbonization temperature has a strong effect on the thermal degradation and the volatilization process of an impregnated sample, leading to an increase in the surface area [25]. The surface areas were determined for the prepared activated carbons at different activation temperatures from 400 °C to 800 °C. The results presented in Figure 3b showed that increasing activation temperatures from 400 °C to 600 °C caused a significant increase in $S_{BET}$, achieving the highest value of 1421 cm$^2$ g$^{-1}$ at 600 °C. Further increases in carbonization temperature led to a decrease in the specific surface area, which can be explained by the shrinkage in the carbon structure, resulting in a reduction in textural properties [14]. Similar trends have also been found for the BET surface area of activated carbon obtained through the $H_3PO_4$ activation of other lignocellulosic precursors [14,26,27].

### 3.1.3. Textural Properties

Keeping in mind that surface development has been strongly connected to the adsorption properties, the more detailed textural results are presented for the selected activated carbon sample obtained with an impregnation ratio of 1.5 and at a carbonization temperature of 600 °C, denoted in further discussion as 1.5/600 (Figure 4 and Table 1). In order to summarize the effect of the optimal impregnation ratio and carbonization temperature on the improvement of the textural and surface characteristics of raw WH, this sample has also been discussed (Figure 4 and Table 1).

Based on the adsorption–desorption isotherm of raw WH (embedded in Figure 4) and the results from Table 1, it can be noticed that the amount of adsorbed nitrogen and the $S_{BET}$ and $V_{tot}$ values are quite low. The raw WH has characteristics of a macroporous material, and the $V_{mic}$, $V_{meso}$, and $D_p$ cannot be measured. The obtained results are expected for raw lignocellulose material. On the other hand, the impregnation process followed by carbonization under optimal conditions dramatically increased the textural properties of the treated sample. The prepared activated carbon show both micro- and mesoporosity. The sample 1.5/600C has a micropore volume of 0.294 cm$^3$ g$^{-1}$ and a significantly higher

mesopore volume, of 0.446 cm$^3$ g$^{-1}$. The percentage of mesoporosity obtained, for the ratio of $V_{meso}$ to $V_{tot}$, is equal to 60.2%. Based on these values, the sample can be described as a dominantly mesopores material, which is a crucial characteristic for the access of the adsorbate molecules to the interior of the adsorbent particles. Several researchers also found that activated carbons derived from different lignocellulose precursors showed micro-and mesoporosity after treatment with phosphoric acid [15,16].

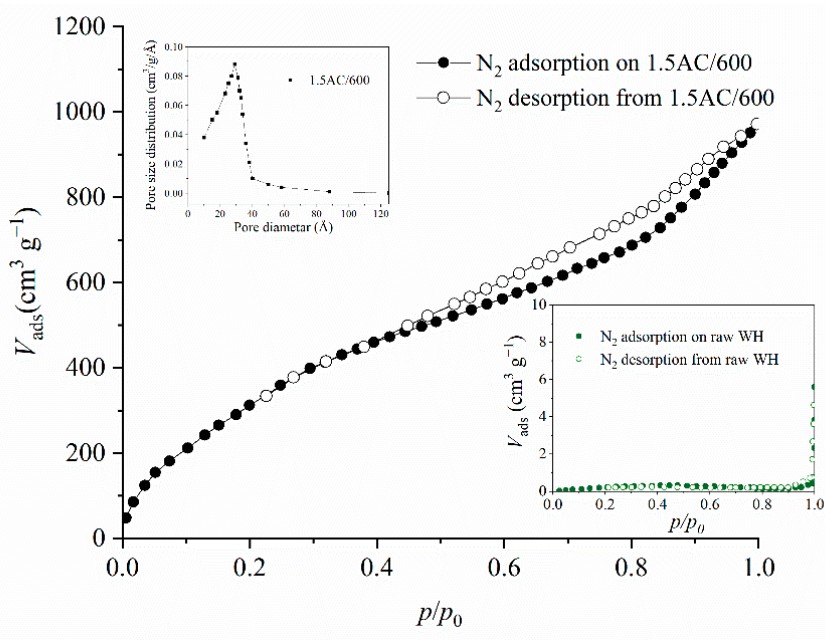

**Figure 4.** The adsorption–desorption isotherms of nitrogen for the sample 1.5/600C.

**Table 1.** Surface area and pore volumes of activated carbon with impregnation ratio 1.5 at 600 °C.

| Sample | $S_{BET}$ (m$^2$ g$^{-1}$) | $V_{tot}$ (cm$^3$ g$^{-1}$) | $V_{mic}$ (cm$^3$ g$^{-1}$) | $V_{meso}$ (cm$^3$ g$^{-1}$) | $D_P$ (nm) |
|---|---|---|---|---|---|
| WH | 2.4 | 0.00872 | - | - | - |
| 1.5/600C | 1421.0 | 0.741 | 0.294 | 0.446 | 2.8 |

According to the literature, the pores can be divided into three different classes: <20 Å micropores, 20 Å < mesopores < 50 nm Å, and >50 Å macropores [28]. The micropores and mesopores, which have been developed mostly during the activation–carbonization process, are most significant to the adsorbing capability.

The pore diameter distribution for 1.5AC/600 mostly lies between 10 Å and 40 Å, which clearly shows that the pore diameter is in the micropore and at the beginning of the mesopore range, with an average pore diameter ($D_p$) of 28 Å.

### 3.1.4. Elemental Analysis

Elemental analysis was performed in order to evaluate the effect of temperature and the applied activation agent on the chemical composition of the selected activated carbon—1.5/600C—in comparison with the starting raw WH material. The elemental analysis of dry WH showed ash, consisting mainly of silica-oxides and metal-oxides, present at 3.94%. The oxygen content was calculated to 47.07% [17]. The result of the elemental analysis of 1.5/600C reveals that the content of carbon, hydrogen, oxygen, nitrogen, and ash was 81.56%, 3.47%, 13.78%, 0.28%, and 0.91%, respectively. During the carbonization process, the carbon content increased, and the major elements present in the investigated sample were carbon and oxygen, which was expected [29]. The elemental analysis shows that content of the major elements is similar to the samples obtained by the activation of WH with ZnCl$_2$ [18].

### 3.1.5. FTIR Analysis

In order to identify the major surface functional groups, the FTIR spectrum of the selected activated carbon—1.5AC/600—was analyzed (Figure 5).

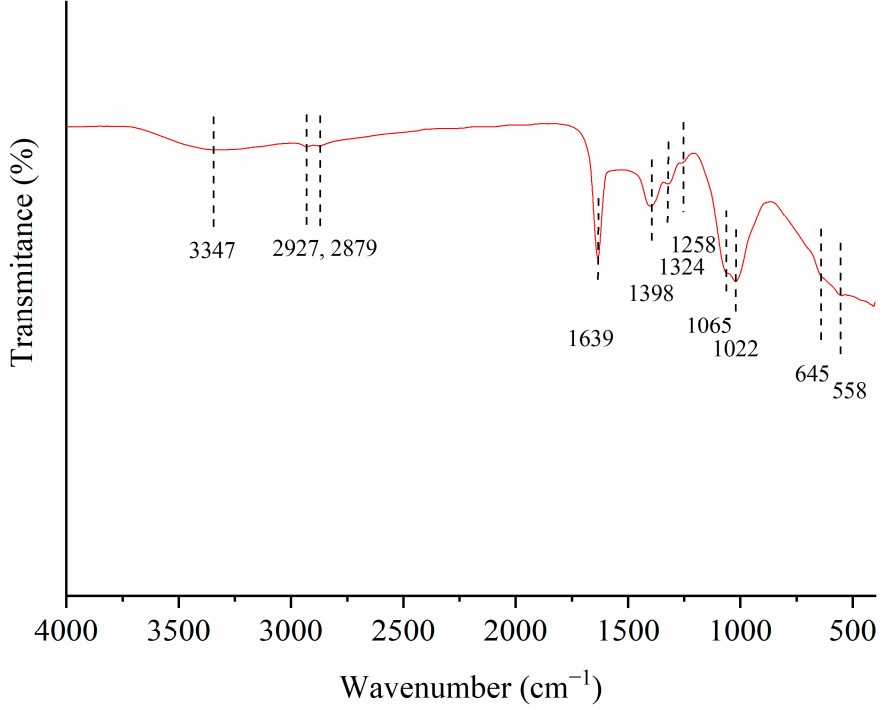

**Figure 5.** The FTIR spectrum of 1.5AC/600.

The assignations of the IR bands from Figure 5 were based on the literature data on infrared spectroscopy absorption [30]. The band at 3347 cm$^{-1}$ is assigned to the O–H stretching vibration of the hydroxyl groups. The two bands at 2927 cm$^{-1}$ and at 2879 cm$^{-1}$ are related to the C–H stretching vibrations of the –CH$_2$– and –CH$_3$ groups, respectively. A sharp and intensive band 1639 cm$^{-1}$ is attributed to both the aromatic C=C bond and/or to the carbonyl group (C=O). The bands at 1398 cm$^{-1}$ and at 1324 cm$^{-1}$ are ascribed to the O-H bending vibration of the carboxylic group and phenol structure, respectively. A weak band at 1258 cm$^{-1}$ can be attributed to the P=O symmetric vibration, while the small shoulder around 1065 cm$^{-1}$ could be connected with the P–O vibration in the acid phosphates and to the symmetrical vibration in the C–O–P chain [31]. The broad band at 1022 cm$^{-1}$ originated from the C–O stretching vibrations, while the two weaker bands between 645 and 558 cm$^{-1}$ were ascribed to the aromatic structures [31,32].

### 3.1.6. Morphological Analysis

Scanning electron microscopy (SEM) was employed to show the morphology of the selected activated carbon, as well as the difference in the surface morphology after treatment of the raw WH material with phosphoric acid and the carbonization process (Figure 6). The SEM image was recorded using a magnification of 3000 times.

The SEM image of the raw WH has been previously published [18]. In the raw WH morphology, some parts of the smooth area and the moderately developed surface can be observed, which have the characteristics of non-treated lignocellulose materials. After the process of chemical activation (by H$_3$PO$_4$) and carbonization (at 600 °C), the morphology of the sample starts to be more developed, with open canals and cracks. Furthermore, the presence of different sizes and shapes of macrostructure pores can be noticed.

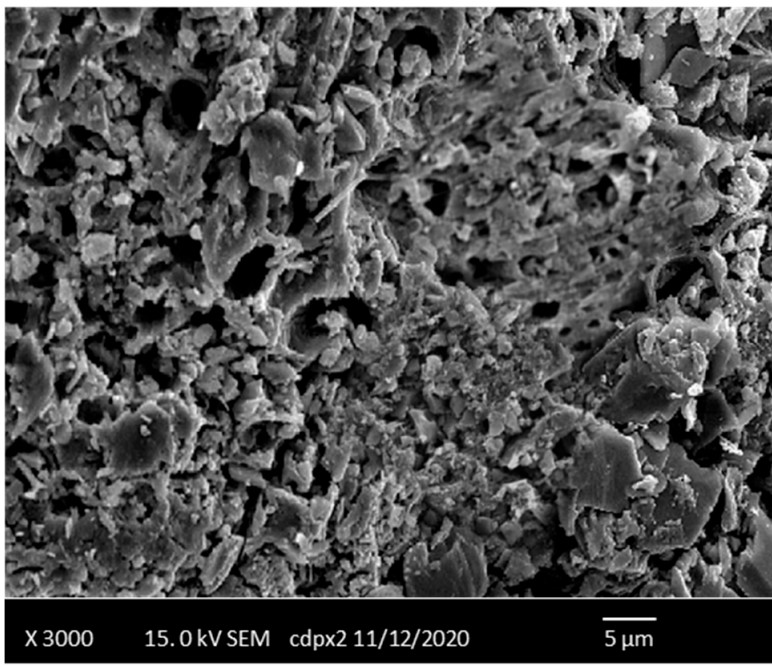

**Figure 6.** The SEM image of the sample 1.5/600C.

### 3.2. Adsorption Study

#### 3.2.1. Effect of the Adsorbent Concentration

The effect of the adsorbent concentration (mg L$^{-1}$) on the amount of the adsorbed MTF on 1.5/600C is presented in Figure 7.

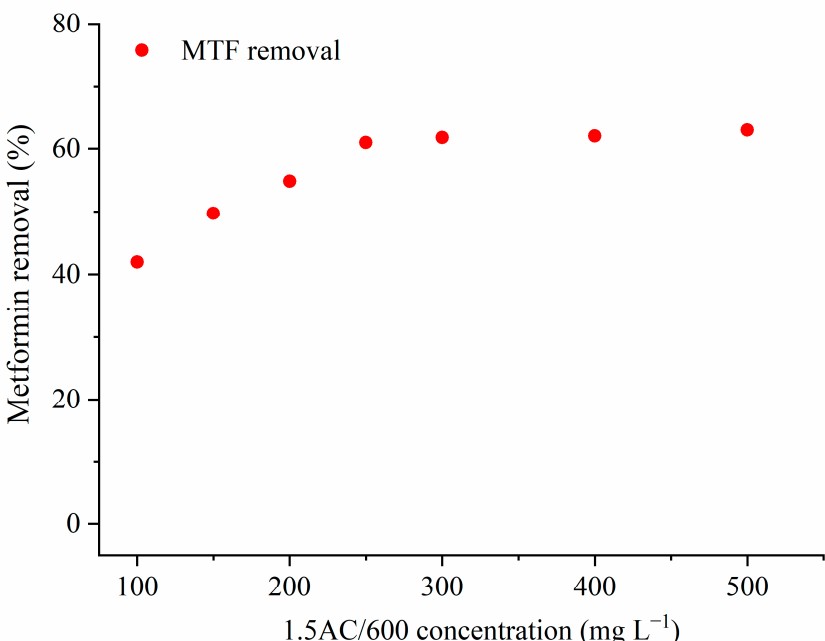

**Figure 7.** The influence of the adsorbent mass on the percentage of metformin removal ($C_{MTF}$ = 20 mg L$^{-1}$; $T$ = 25 °C; $t$ = 120 min).

The results show that the percentage of MTF removal increased from 41.9% to 61.0%, with an increase in AC concentration from 100 mg L$^{-1}$ to 250 mg L$^{-1}$. The AC concentrations higher than 250 mg L$^{-1}$ did not significantly affect the removal of MTF. Therefore, a concentration of adsorbent of 250 mg L$^{-1}$ was chosen for further adsorption experiments. The higher concentrations of adsorbent can lead to the aggregation of the adsorbent parti-

cles, decreasing the available surface area and, consequently, decreasing the removal of the adsorbate [33].

### 3.2.2. Influence of the Metformin Concentration

The influence of the initial adsorbate concentration on the amount of the adsorbed MTF for different contact times is presented in Figure 8.

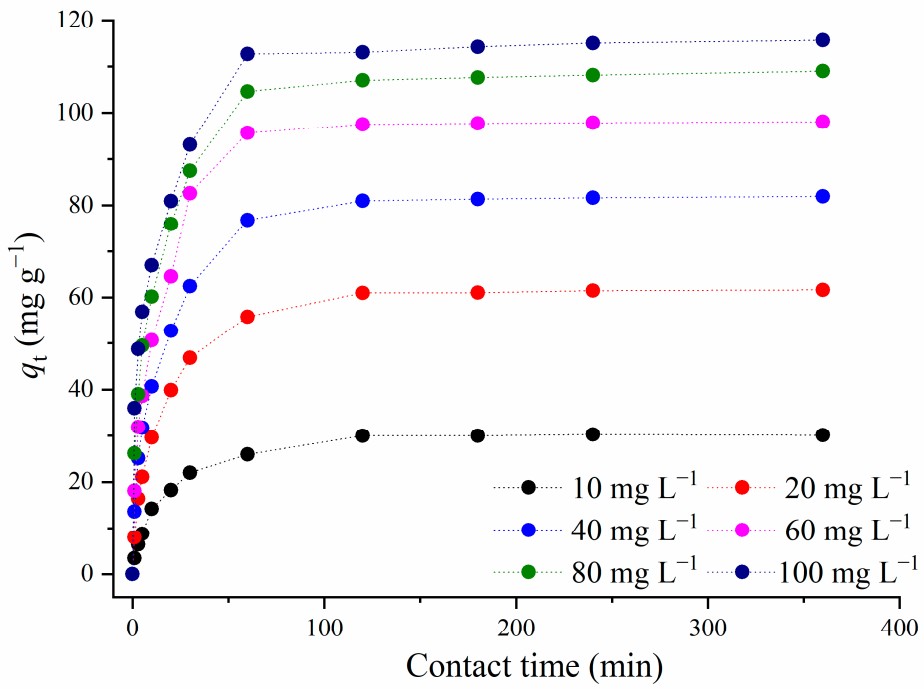

**Figure 8.** The influence of the initial MTF concentration on the amount of the adsorbed drug over time $t$ ($C_{1.5AC/600}$ = 250 mg L$^{-1}$; $T$ = 25 °C).

Figure 8 shows that the MTF adsorption approached a plateau after 120 min, for all investigated initial concentrations of the adsorbate. Due to this result, 120 min was taken as the equilibrium adsorption time. For the initial concentrations of MTF higher than 40 mg L$^{-1}$, more than 50% of MTF was adsorbed within the first 15 min, while for the lower concentrations of adsorbate (10 mg L$^{-1}$ and 20 mg L$^{-1}$), half of the totally adsorbed metformin was removed in 20 min. The drastic increase in adsorption for such short times may be due to the availability of vacant adsorption sites on the adsorbate surface. With increasing contact time, adsorption sites become occupied, and adsorption equilibrium is reached. After an equilibrium time of 120 min, there is no significant increase in the amount of adsorbed MTF.

### 3.2.3. Effect of the pH of Metformin Solution

The pH of the adsorbate solution is known to be an essential parameter that affects adsorption behavior. The pH of the adsorbate solution affects the adsorbent surface charge and surface ionization of the adsorbate material [34]. The effect of the initial pH of the metformin solution on the percentage removal of MTF was studied by varying the initial pH under constant process parameters. The results of the MTF adsorption under a range of pH solutions from 2–12 are shown in Figure 9, along with the point of zero charges for 1.5AC/600.

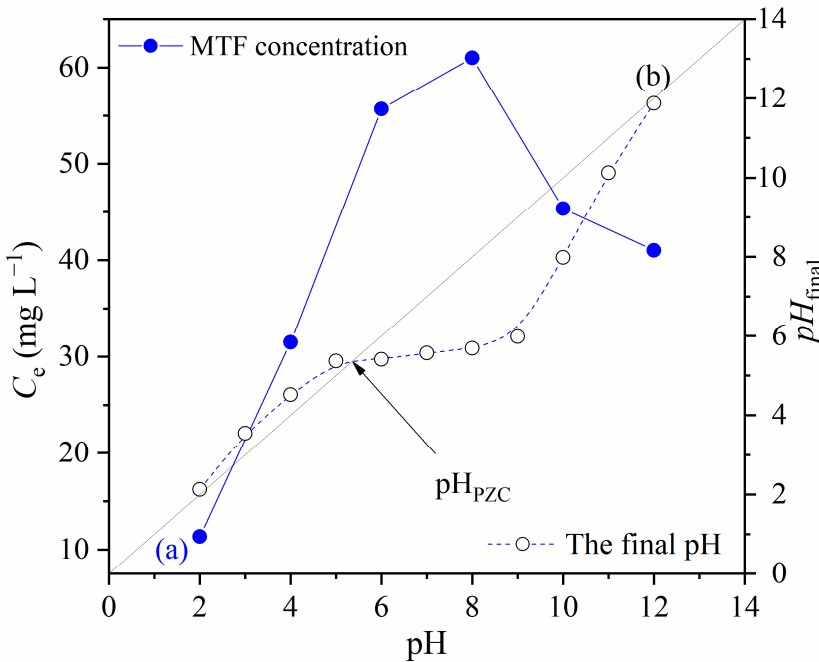

**Figure 9.** (a) The effect of the pH of MTF solution on adsorption; (b) the point of zero charge for 1.5/AC600; ($C_{MTF}$ = 20 mg L$^{-1}$; $C_{1.5AC/600}$ = 250 mg L$^{-1}$; $T$ = 25 °C; $t$ = 120 min).

From Figure 9, it was observed that the uptake of MTF increased when the pH value changed from 2.0 to 8.0 and then decreased when the pH increased from 8.0 to 12.0. These phenomena can be attributed to the surface charge of 1.5AC/600 and the speciation of metformin at the different pH values. The point of zero charge (Figure 9) was determined to be at pH = 5.35. The surface of 1.5AC/600 below this value is positively charged and negatively charged above the pH$_{PZC}$ value.

At a pH below a p$K_{a1}$ of 2.8, MTF is mainly present as a biprotonated species, and, at the same time, the surface of the adsorbent is positively charged. Therefore, repulsive interactions between adsorbent and adsorbate occurred, and the lowest amount of adsorbed MTF (15.10 mg L$^{-1}$) is observed at pH = 2.0. Further, at 4.0 < pH < 10.0, MTF is mainly present in a monoprotonated form, while the surface of the adsorbent starts to be negatively charged above 5.35. Hence, the significant increase in adsorbed MTF can be noticed at pH = 6.0, with a maximum of adsorbed MTF at pH = 8.0. This adsorption behavior has been caused by the attractive forces between the adsorbent surface and MTF. Finally, at pH values above the p$K_{a2}$ of 11.6, MTF is present in molecular form, and adsorption only through the pore diffusion mechanism can be present [35].

It is important to point out that the native pH of the metformin hydrochloride solution was 6.8. At this pH value, the percentage of removed MTF was 49.5%, which is 96% of the amount of adsorbed MTF at the optimum pH. This finding favors the application of 1.5AC/600 in real systems, since the adsorption process does not require the addition of extra acid or base to achieve an amount of adsorbed metformin almost equal to that obtained for optimal adsorption conditions.

### 3.3. Modeling of the Experimental Adsorption Data

In order to describe the nature of the MTF adsorption at 1.5AC/600, various isotherm and kinetics models were applied to the adsorption data. Adsorption isotherms can provide information related to the interaction between adsorbate and adsorbent and optimize the use of adsorbent. Therefore, the adsorption results are fitted by the Langmuir [36], Freundlich [37], and Redlich–Peterson isotherms [38,39]. On the other hand, the theoretical kinetics model investigated the mechanism of adsorption and potential rate-controlling steps, including the mass transport and chemical reaction processes. Hence, the experimen-

tal kinetic data were fitted with the pseudo-first kinetic model [40], pseudo-second-order kinetics' model [41], and Weber–Morris intra-particle diffusion model [42].

### 3.3.1. Adsorption Isotherms

Three different adsorption isotherm models, Langmuir, Freundlich, and Redlich–Peterson, are applied to the experimental data for MTF adsorption at 1.5AC/600 and 25 °C, which are presented in Figure 10, while the calculated isotherms parameters are given in Table 2.

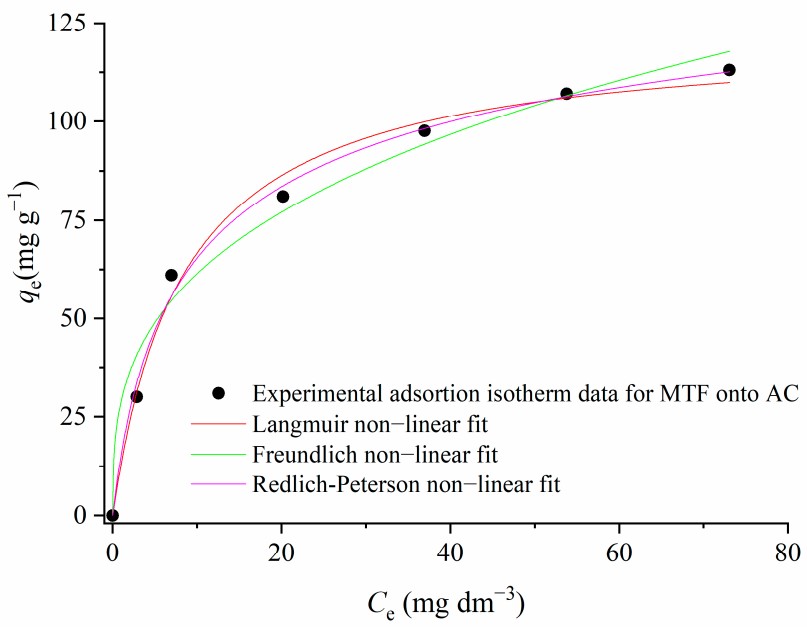

**Figure 10.** Adsorption isotherm of metformin onto 1.5AC/600 ($C_{MTF}$ = 10–100 mg L$^{-1}$; $C_{1.5AC/600}$ = 250 mg L$^{-1}$, $T$ = 25 °C, $t_{eq}$ = 120 min).

**Table 2.** Calculated isotherm parameters for applied isotherm models for MTF adsorption onto 1.5AC/600.

| Isotherm Model | Adsorption Models Parameters | | | |
|---|---|---|---|---|
| Langmuir | $K_L$ (dm$^3$ mg$^{-1}$) | $R_L$ | $q_{max}$ (mg g$^{-1}$) | $R^2$ |
| | 0.1194 | 0.457 | 122.47 | 0.997 |
| Freundlich | $K_F$ (mg g$^{-1}$) (dm$^3$ mg$^{-1}$)$^{1/n}$ | | nF | $R^2$ |
| | 28.871 | | 3.05 | 0.977 |
| Redlich–Peterson | $K_R$ (L g$^{-1}$) | $A$ (L mg$^{-1}$)$^n$ | $n_R$ (g mg$^{-1}$) | $R^2$ |
| | 19.163 | 0.246 | 0.894 | 0.995 |

$K_L$—Langmuir constant; $R_L$—Langmuir factor related with favorable adsorption; $q_{max}$—adsorption capacity, $R^2$—coefficient of determination; $K_F$—Freundlich constant; $n_F$—Freundlich factor related to heterogeneity of adsorption; $K_R$—Redlich–Peterson isotherm constants, $A$—Redlich–Peterson parameter; $n_R$—the exponent, which lies between 1 and 0.

Based on the values of the coefficient of determination—$R^2$ (Table 2)—obtained for each investigated adsorption model, the MTF adsorption could be best described by the Langmuir isotherm, followed by the Redlich–Peterson model, while the Freundlich model seems to be less suitable.

Langmuir's adsorption model can describe the MTF adsorption process as mainly homogeneous at energetically equal adsorption sites, where adsorbate molecules form the monolayer at the adsorbent surface. For a Langmuir-type adsorption process, the isotherm shape can be classified by a dimensionless constant separation factor $R_L$. The value of $R_L$ indicates the shape of the isotherms to be either unfavorable ($R_L > 1$), linear ($R_L = 1$), favorable ($0 < R_L < 1$), or irreversible ($R_L = 0$). For the investigated adsorption process of

MTF on 1.5AC/600, the calculated $R_L$ has the value of 0.457 (Table 2); therefore, the value is in the range between 0 and 1, and the adsorption of MTF on 1.5AC/600 can be regarded as a favorable process. Further, the Langmuir model predicts an adsorption maximum of $q_{max} = 122.47$ mg g$^{-1}$, classifying the investigated activated carbon as a very efficient one. The Redlich–Peterson model has also shown very good agreement with the adsorption data, with a coefficient of correlation of 0.995. The Redlich–Peterson isotherm describes the adsorption equilibrium over a wide concentration range and can be applied in either homogeneous or heterogeneous systems due to its versatility. The good agreement of experimental results with the Redlich–Peterson adsorption isotherm was also found for dyes' adsorption on mesoporous activated carbon prepared from pistachio shells with NaOH activation [43].

The literature review of $q_{max}$ values for metformin adsorptions on different activated carbons derived from various types of biomasses, along with other adsorbents, is presented in Table 3.

**Table 3.** Comparison of metformin adsorption capacity of different adsorbents.

| Adsorbent | Adsorption Parameters | $q_{max}$ (mg g$^{-1}$) | References |
|---|---|---|---|
| 1.5AC/600 | $S_{BET} = 1421$ m$^2$ g$^{-1}$; pH = 8.20; $T = 25$ °C, $C_{MTF} = 10–100$ mg dm$^{-3}$ | 122.47 | Current study |
| Multi-walled carbon nanotubes, commercial | $S_{BET} = 250–280$ m$^2$ g$^{-1}$; $T = 295$ K, $C_{MTF} = 10–88$ mg dm$^{-3}$ | 79.94 | [44] |
| Granular activated carbon | $S_{BET} = 1500$ m$^2$ g$^{-1}$; $T = 295$ K; $C_{MTF} = 10–88$ mg dm$^{-3}$ | 72.56 | [44] |
| Graphene oxide | pH = 6; $C_{MTF} = 8–40$ mg L$^{-1}$, contact time 160 min; $T = 288, 303$ and 318 K. | 96.748 89.099 88.517 | [1] |
| Activated carbon from agricultural waste | $T = 20$ °C, $C_{MTF} = 10–200$ mg L$^{-1}$ pH = 7, contact time 125 min. | 44.84 | [45] |
| Sibipiruna activated carbon | $C_{MTF} = 500$ mg L$^{-1}$, $T = 30$ °C, pH = 13, contact time 360 min. | 248.48 | [35] |
| Water-treated clay Acid-treated clay | $S_{BET} = 9.5 –11.5$ m$^2$ g$^{-1}$; pH = 6, $C_{MTF} = 1–20$ mg L$^{-1}$ $T = 298$ K; contact time 30 min. | 25.268 33.788 | [46] |
| Fe-ZSM-5 nano-adsorbent | $T = 25$ °C, $C_{MTF} = 5–20$ mg L$^{-1}$, contact time 20 min. | 14.992 | [47] |
| Activated carbon from orange peel | $T = 323$ K, pH = 7; contact time 240 min. | 50.99 | [48] |
| Hydrogen peroxide modified biochar | $T = 308.15$ K, $C_{MTF} = 0.05–3.6$ mmol L$^{-1}$, contact time 24 h. | 107.33 | [49] |

Based on the presented literature review, the predicted $q_{max}$ value for metformin adsorption in this study is higher than the majority of the $q_{max}$ values obtained in the literature for metformin adsorption on various adsorbents. Therefore, the activated carbon obtained in the present study can be considered for testing in real systems.

### 3.3.2. Kinetics of the Adsorption

The kinetic models were applied to the experimental adsorption results for MTF adsorption on 1.5AC/600, in order to investigate the mechanism of the adsorption and potential rate-controlling steps, including the mass transport and chemical reaction processes. The linear plots of the pseudo-first-order kinetic model, pseudo-second-order

kinetic model, and Weber–Morris intra-particle diffusion model are given in Figure 11a–c, while the corresponding calculated kinetic parameters are presented in Table 4.

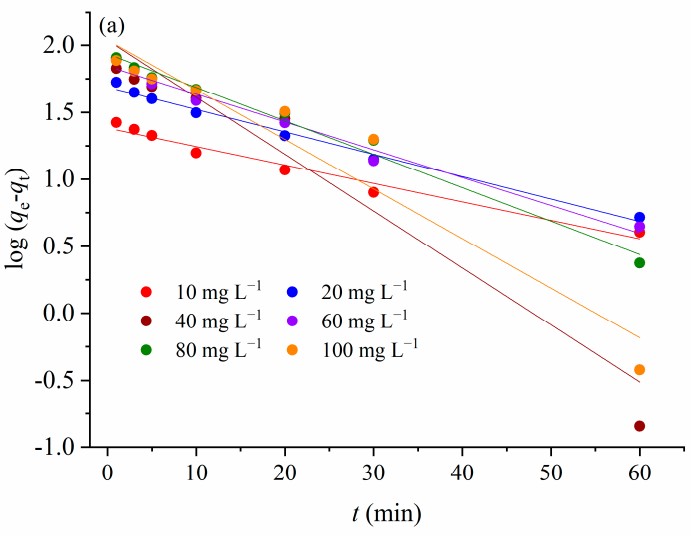

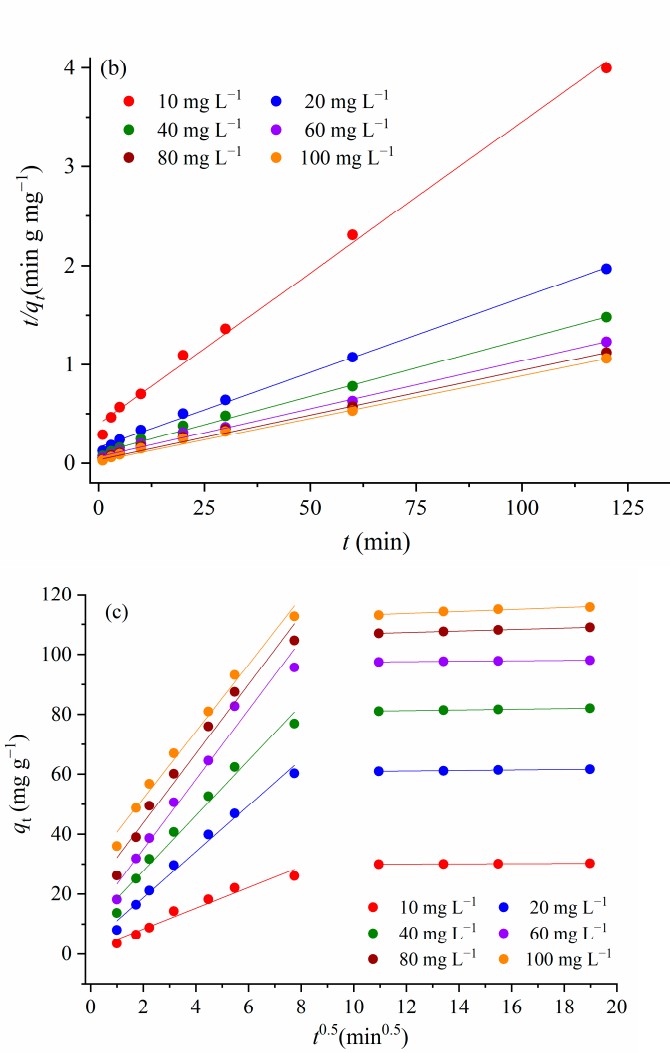

**Figure 11.** (**a**) Pseudo-first order kinetics model, (**b**) pseudo-second order kinetics model, and (**c**) Weber–Morris intra-particle diffusion model.

**Table 4.** The parameters calculated for pseudo-first order kinetic model, pseudo–second order kinetic model, and Weber–Morris intra-particle diffusion model for adsorption of MTF on 1.5AC/600.

| Model Parameters | $C_0$ (mg g$^{-1}$) | | | | | |
|---|---|---|---|---|---|---|
| | 10 | 20 | 40 | 60 | 80 | 100 |
| $q_e{}^{exp}$ (mg g$^{-1}$) | 30.01 | 60.99 | 81.02 | 97.52 | 107.00 | 113.01 |
| Pseudo-first order kinetics model | | | | | | |
| $q_e$ (mg g$^{-1}$) | 24.11 | 48.76 | 59.34 | 69.77 | 86.22 | 109.19 |
| $k_1$ $10^{-3}$ (min$^{-1}$) | 5.99 | 7.25 | 8.42 | 9.03 | 10.9 | 16.4 |
| $R^2$ | 0.972 | 0.991 | 0.899 | 0.983 | 0.987 | 0.929 |
| Pseudo-second order kinetics model | | | | | | |
| $q_e$ (mg g$^{-1}$) | 32.79 | 65.79 | 86.96 | 104.17 | 111.11 | 119.05 |
| $k_2 \cdot 10^{-3}$ (g mg$^{-1}$ min) | 2.34 | 1.44 | 1.24 | 1.18 | 1.35 | 1.45 |
| $R^2$ | 0.996 | 0.998 | 0.996 | 0.996 | 0.997 | 0.996 |
| Weber–Morris intra-particle diffusion model | | | | | | |
| $C_{id1}$ | 1.40 | 3.37 | 9.26 | 11.172 | 11.57 | 11.68 |
| $k_{WM1}$ (mg g$^{-1}$ min$^{-1/2}$) | 3.46 | 7.69 | 9.23 | 11.64 | 20.56 | 29.536 |
| $R^2$ | 0.962 | 0.987 | 0.980 | 0.974 | 0.976 | 0.988 |
| $k_{WM2}$ (mg g$^{-1}$ min$^{-1/2}$) | 0.031 | 0.086 | 0.119 | 0.069 | 0.247 | 0.327 |
| $R^2$ | 0.985 | 0.969 | 0.980 | 0.994 | 0.999 | 0.946 |

$q_e{}^{exp}$—experimentally obtained value of the amount of the adsorbed MTF at equilibrium; $q_e{}^{cal}$—calculated value of the amount of the adsorbed MTF, based on appropriate kinetic model; $k_1$—pseudo first order constant; $k_2$—pseudo-second order constant; $R^2$—coefficient of determination; $k_{WM1}$, $k_{WM2}$—Weber–Morris intraparticle diffusion rate constants; $C_{id}$—intercept of the linear plot, corresponding to initial adsorption.

Based on the values of the kinetics parameters and correlation coefficient presented in Table 4, the adsorption kinetics of MTF adsorption for 1.5AC/600 are the best fitting with the pseudo-second-order (PSO) kinetic model. The PSO kinetic model has usually been associated with the surface-reaction kinetic step, controlling the adsorption rate in solid/solution systems [50]. There are accepted assumptions that the adsorption rate of the ion exchange reaction of the systems described by this model is responsible for the removal kinetics. According to the literature, the constant—$k_2$—of the adsorption systems described by the PSO kinetic showed that it can be both dependent on and independent of the initial concentration of the adsorbate. The adsorption of MTF onto 1.5/AC600 does not show the correlation between $k_2$ and the initial MTF concentration. This finding is in agreement with similar adsorption systems, where activated carbons were used as adsorbents [50–52].

Weber–Morris's equation was used to describe the intra-particle diffusion (Figure 11c). If the regression of $q_t$ versus $t^{1/2}$ is linear and passes through the origin, then intra-particle diffusion is the only rate-limiting step [53]. However, for adsorption of the MTF in the investigated concentration range onto 1.5AC/600, multi-linearities are observed (Figure 11c), where each linear segment represents a controlling mechanism or several simultaneous controlling mechanisms during the adsorption process. The first sharper portion is attributed to the rapid adsorption on the external surface and the current interaction between MTF and the available sites on the adsorbent surface. The slopes of the first linear portion do not pass through the origin, indicating that intraparticle diffusion is not the rate-limiting step. Based on the values of $C_{id}$ (Table 4), it can be concluded that the effective role of the boundary layer on the adsorption rate constant increased with an increase in the MTF concentration in the solution. This result is in agreement with the literature data, e.g., Raji and Pakizeh [54] found that a higher bulk liquid concentration of the adsorbate led to an increase in both the rate constant and value of the boundary layer thickness ($C_{id}$).

The second portion describes the gradual layer adsorption stage, where intraparticle diffusion is rate limiting.

### 3.4. Thermodynamics of the Adsorption

The thermodynamic parameters for the adsorption process can be calculated from the following equation:

$$\Delta G = -R \times T \times \ln K_d \tag{5}$$

where $R$ is the gas constant of 8.314 J K$^{-1}$·mol$^{-1}$, $K_d$ is the equilibrium constant, and $T$ (K) is the temperature. The $K_d$ value is calculated from the following equation:

$$K_d = q_e / C_e \tag{6}$$

where $q_e$ is the amount of MTF adsorbed at equilibrium (mg g$^{-1}$), and $C_e$ is the equilibrium concentration of MTF in the solution (mg L$^{-1}$). The standard enthalpy ($\Delta H$) and entropy ($\Delta S$) of adsorption can be estimated from the Van 't Hoff equation:

$$\ln K_C = -\Delta H / RT + \Delta S / R \tag{7}$$

The results of the thermodynamic study are presented in Figure 12.

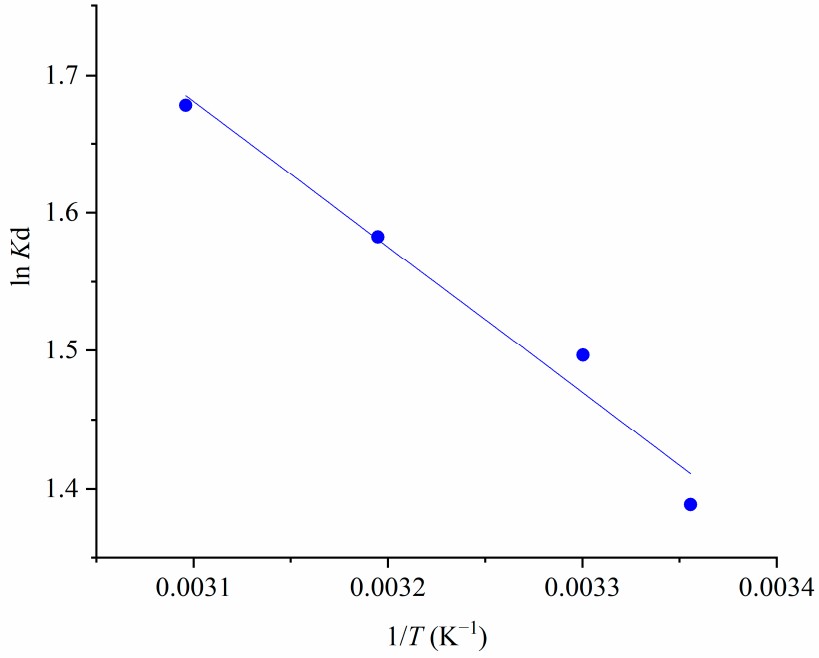

**Figure 12.** The Van 't Hoff plot for MTF adsorption on 1.5AC/600 ($C_{MTF}$ = 40 mg L$^{-1}$; $C_{1.5AC/600}$ = 250 mg L$^{-1}$; $T$ = 25–50°C/298–323 K; $t$ = 120 min).

Based on the values of the slope and intercept, the values of $\Delta H$ and $\Delta S$ were calculated, with values of 8.77 kJ mol$^{-1}$ and 41.14 J·K$^{-1}$·mol$^{-1}$, respectively. The standard Gibbs free energy change ($\Delta G°$) was found to be −3.44 kJ mol$^{-1}$ at 298 K, confirming the feasibility and spontaneity of the adsorption process.

### 3.5. Desorption and Reusability Study

The reusability of 1.5AC/600 can be one of the most significant criteria for its practical application in real systems. Hence, finding the appropriate and most effective desorption agent was one of the most important tasks of this study. The desorption was primarily performed from 0.1 M solutions of NaOH, HCl, and NaCl (Figure 13).

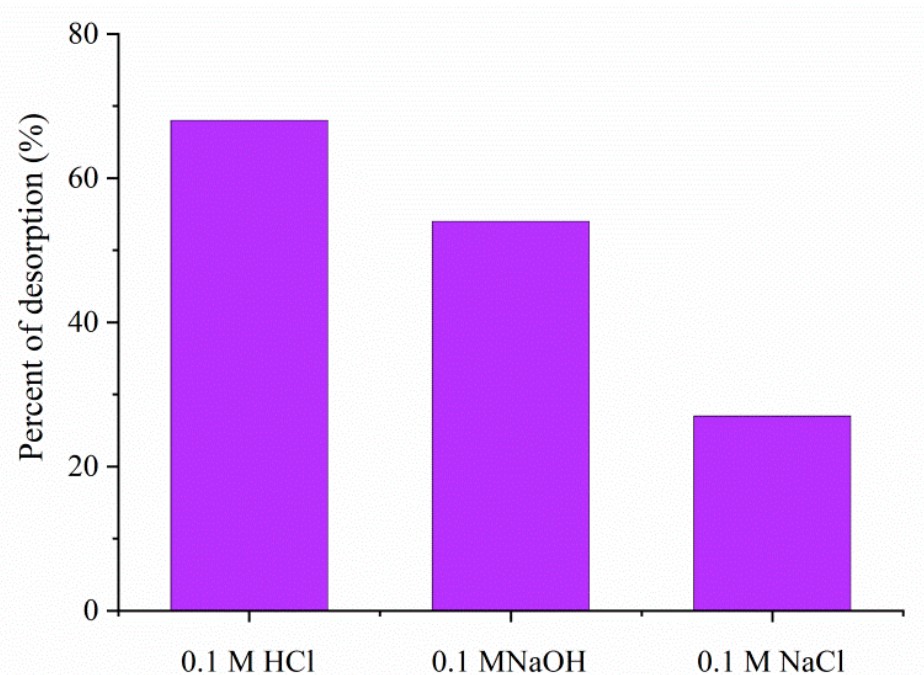

**Figure 13.** Desorption of MTF from MTF-saturated 1.5AC/600 in 0.1 M solutions of HCl, NaOH, and NaCl.

The experimental data indicate that in the presence of HCl, about 68% of adsorbed MTF is released. In the presence of NaOH and NaCl desorption, the percentages are found to be 54% and 27%, respectively. The obtained result for the desorption study is in agreement with the literature data. Several studies dealing with desorption found that the the hydroxyl and carboxylic groups present on the surface of the adsorbent made it susceptible to easy desorption and regeneration [55,56].

Based on the literature survey, it was found that the desorption process for some adsorption systems is efficient in the presence of an acid–ethanol mixture [57,58]. In order to achieve a higher amount of the desorbed MTF than that obtained in the presence of 0.1 M HCl, mixtures of 0.1 HCl and 96% ethanol in different volume ratios (3:1, 2:1, 1:1, 1:2, 1:3) were tested and are shown in Figure 14.

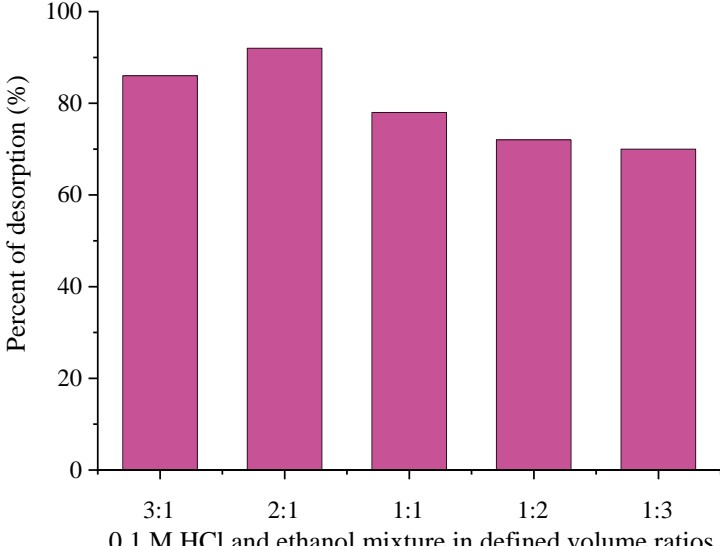

**Figure 14.** Desorption of MTF from 0.1 M HCl/ethanol mixture.

Based on the results from Figure 14, it was obvious that the presence of ethanol in mixture with HCl significantly contributed to MTF desorption. The amounts of desorbed MTF from all investigated HCl/ethanol solutions were higher than the amount of desorbed MTF in the presence of 0.1 M HCl. The MTF desorption efficiency reached a maximum of 92%, when a desorption solution of 0.1M HCl/ethanol in volume ratio 2:1 was used, while a further increase of the ethanol volume led to a decrease in the desorption efficiency. This finding indicated that adsorption of MTF onto 1.5AC/600 occurred via different interactions (electrostatic interaction, hydrogen bonding, etc.). For further desorption experiments and testing of the reusability of 1.5AC/600, a mixture of HCl/ethanol with a volume ratio 2:1 was chosen as the desorption agent.

The reusability of 1.5AC/600 was tested in five successive cycles—Figure 15.

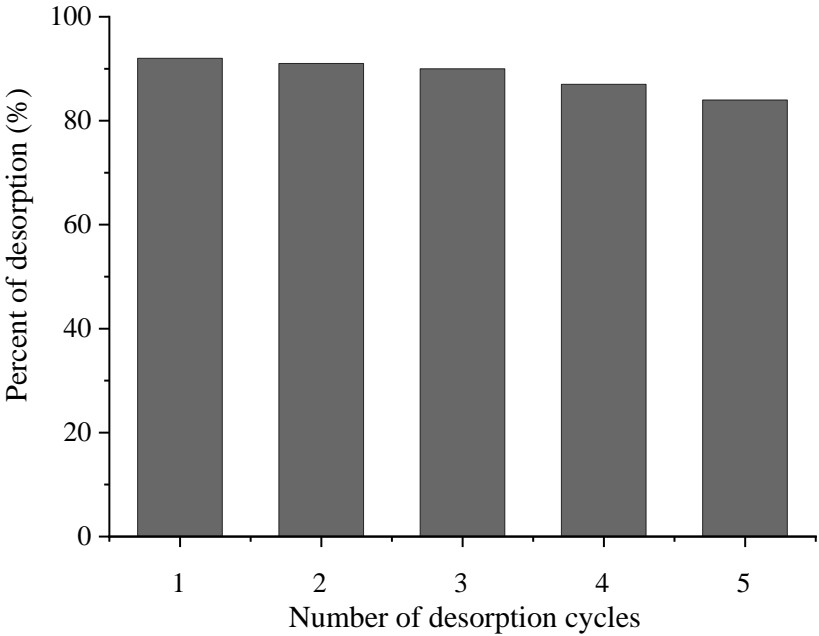

**Figure 15.** Desorption of MTF in five consecutive adsorption–desorption cycles.

Desorption experiments in the presence of optimal desorption solution 0.1 M HCl/ethanol in a vol. ratio 2:1 show that the 1.5AC/600 could be regenerated without a significant loss in its initial efficiencies after five adsorption–desorption cycles, when the percentage of the desorbed MTF is 84%. These results demonstrated that 1.5AC/600 exhibits good reusability and stability to remove MTF from pharmaceutical wastewaters and, potentially, could be implemented as an effective adsorbent material in real systems.

## 4. Conclusions

Water hyacinth biomass was used as precursor lignocelluloses material for the production of activated carbons in the presence of $H_3PO_4$ as a chemical activator. The investigated impregnation ratios between $H_3PO_4$ and dry water hyacinth were in the range of 0.5–3.0, while the applied carbonization temperatures were 400 °C, 500 °C, 600 °C, 700 °C, and 800 °C. The yield and specific surface area of the obtained activated carbons strongly depend of the investigated parameters of synthesis, i.e., the impregnation ratio and carbonization temperature. The best surface development ($S_{BET}$ = 1421 m$^2$ g$^{-1}$) shows a sample with an impregnation ratio of 1.5 and a carbonization temperature of 600 °C. This activated carbon with the best textural properties was further used as an adsorbent for metformin, an anti-diabetic drug, which has already entered the environment.

Adsorption data showed that the adsorption takes place quickly, and the equilibrium time was estimated at 120 min. The pH of the metformin solution had a strong impact on the amount of adsorbed pharmaceutics, with an optimal pH value of 8. The percentage of the adsorbed metformin at native pH was close to that obtained at optimal pH. This

finding revealed that the application of 1.5AC/600 in real systems does not require the addition of an extra acid or base for achieving the optimal adsorption conditions.

The adsorption isotherms were fitted with Langmuir, Freundlich, and Redlich–Peterson isotherm models, and the Langmuir model showed the best fit, describing metformin adsorption as one that is in a monolayer form on energetically equal and homogenously distributed adsorption sites. The important adsorption parameter—Langmuir adsorption capacity—has a value of $q_{max}$ = 122.47 mg g$^{-1}$, classified the selected adsorbent as one of the most efficient for metformin removal in the literature. The pseudo-first, pseudo-second, and Weber–Morris intraparticle diffusion models have been employed to describe adsorption kinetics of metformin onto 1.5AC/600. The kinetic modeling revealed that metformin adsorption follows the pseudo-second-order kinetics, which indicates a possible chemisorption mechanism, specifically an ion-exchange mechanism. According to the Weber–Morris intra-particle diffusion model, intra-particle diffusion is not the rate-limiting step. On the other hand, the boundary layer has an effective role in the adsorption rate constant, which increased with an increase in the initial metformin concentration. The adsorption of metformin was performed in the temperature range from 25–50 °C, and thermodynamic interpretation suggests that the investigated adsorption process is spontaneous and endothermic, with values for the free Gyps energy ($\Delta G°$) and $\Delta H°$ of −3.44 kJ mol$^{-1}$ and 8.77 kJ mol$^{-1}$, respectively. It was found that the highest amount of desorbed metformin was achieved in the presence of an HCl/ethanol mixture, at a volume ratio of 2:1. It was found that the percentage of regeneration of 1.5AC/600 after the fifth consecutive adsorption–desorption cycle was 84%. Therefore, the selected adsorbent can be used several times without significant adsorption capacity abatement and can be potentially applied in real wastewater systems.

**Author Contributions:** Conceptualization, A.H.M. and M.K.; methodology and investigation, A.H.M.; experimental procedures, A.H.M.; analysis, A.H.M., I.R. and M.K.; writing—original draft preparation, A.H.M.; writing—review and editing, A.H.M., M.K., I.R. and M.I. All authors have read and agreed to the published version of the manuscript.

**Funding:** This work was supported by the Ministry of Education, Science and Technological Development of the Republic of Serbia (Contract No. 451-03-68/2022-14/200135).

**Institutional Review Board Statement:** Not applicable.

**Informed Consent Statement:** Not applicable.

**Data Availability Statement:** Not applicable.

**Conflicts of Interest:** The authors declare no conflict of interest.

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
