# Peer review of "Adsorption of Metformin on Activated Carbon Produced from the Water Hyacinth Biowaste Using H3PO4 as a Chemical Activator"

_sustainability, doi:10.3390/su141811144_

Round 1

Reviewer 1 Report

In this work, the authors investigates the effect of the various impregnation weight ratios of H3PO4 and dry hyacinth, as well as different carbonization temperatures on surface characteristics of the produced activated carbons. The research is very interesting, but there are problems.

1. The characterization is not particularly comprehensive, such as the lack of dry hyacinth and other characterization.

2. A schematic diagram of the preparation process is preferred.

3. It is suggested that pictures such as Figure 2 and figure 8 be merged into one. And the format of the picture as standard as possible.

Reviewer 2 Report

This paper described the fabrication of activated carbon from water hyacinth biowaste, which was then used as adsorbents for removing metformin from water. Overall, the manuscript is well organized and the discussion is convincing. I suggest its acceptance after a minor revision.

1. To show the advantage of H3PO4 activator, the application of other types of chemicals as activators should be compared, such as ZnCl2 and KOH. 

2. Since the regeneration performance is not good enough, more experimental work should be carried out to  realize the  repeatedly adsorption for several cycles.

Reviewer 3 Report

Title: Adsorption of metformin on activated carbon produced from the water hyacinth biowaste using H3PO4 as a chemical activator

After reviewing the present manuscript, I found that the authors made interesting work and all required analysis for the Adsorption of metformin on activated carbon produced from the water hyacinth biowaste using H3PO4 as a chemical activator.
I found that this manuscript is very fit with the Sustainability MDPI and need minor revision for publication.

General observations - Manuscript may not have been proof-read sufficiently before the submission, and author comments were found in several places. Please remove them before submitting the revised version. Some formatting issues were observed.

Use indent when you start a new paragraph. Be consistent with the space between units and numbers. Use of stops in appropriate places and correct way of referencing must be carefully looked.

1)      Graphical abstract not included- Adding a graphical abstract would help add weightage to the manuscript.

2)      Include highlights to the manuscript, to present the most significant findings of the research.

3)      Keywords: Use caps for each new word.

4)      Line 56- Rephrase the sentence “industrial and agricultural side products” to “industrial and agricultural byproducts”.

5)      Line 95- You need to elaborate the procedure for the preparation of raw WH rather than just citing?

6)      Line 213- Indent

7)      Line 219- Indent

8)      Line 257- Indent

9)      Line 298- Rearrange subheading number “3.1.5. Elemental analysis”

10)  Line 311- Rearrange subheading number “3.1.6. FTIR analysis”

11)  Line 313- Indent

12)  Line 319- Indent

13)  Line 332- Rearrange subheading number “3.1.7. Morphological analysis”

14)  Line 334- Indent

15)  Can you compare with figures, all the performed characterizations of phosphoric acid treated WH with raw WH?  

16)  Line 368- Indent

17)  Line 386- Indent

18)  Line 470- Table-3: You can you add more adsorbents to the comparison table

19)  References need to be arranged in alphabetical order

Round 2

Reviewer 1 Report

The authors have carefully revised the relevant issues and are ready for publication.

Author Response

Revision is based on the comment of the reviewer 1 “English language and style are fine/minor spell check required”. Manuscript has been carefully checked and the corresponding corrections were incorporated in the new version of our manuscript (labelled R2).